# Implementing a Novel Facility-Community Intervention for Strengthening Integration of Infant Nutrition and Family Planning in Mara and Kagera, Tanzania

**DOI:** 10.3390/ijerph18084105

**Published:** 2021-04-13

**Authors:** Chelsea M. Cooper, Mary Drake, Justine A. Kavle, Joyce Nyoni, Ruth Lemwayi, Lemmy Mabuga, Anne Pfitzer, Mary Makungu, Elizabeth Massawe, John George

**Affiliations:** 1USAID’s Maternal and Child Survival Program/Jhpiego, Washington, DC 20036, USA; cmcoope@gmail.com (C.M.C.); anne.pfitzer@jhpiego.org (A.P.); 2Jhpiego, Baltimore, MD 21231, USA; 3University Medical Centre Groningen, Department of Health Sciences, Global Health, University of Groningen, 9713 GZ Groningen, The Netherlands; 4Kavle Consulting, LLC, Washington, DC 20001, USA; justine@kavleconsulting.com; 5Institute of Social Work, University of Dar es Salaam, Dar es Salaam, Tanzania; jnyoni@gmail.com; 6Jhpiego, Dar es Salaam, Tanzania; ruth.lemwayi@jhpiego.org; 7USAID’s Maternal and Child Survival Program/Jhpiego, Dar es Salaam, Tanzania; lmedard2011@gmail.com; 8Government of Tanzania, Musoma, Mara Region, Tanzania; makungum@ymail.com; 9Bukoba Referral Hospital, Government of Tanzania, Bukoba, Kagera Region, Tanzania; massaweelizabeth90@yahoo.com; 10UNICEF, Dar es Salaam, Tanzania; georgejn2000@yahoo.com

**Keywords:** exclusive breastfeeding, family planning, postpartum family planning, lactational amenorrhea method, maternal infant nutrition, service integration, community health, male engagement, implementation science

## Abstract

Tanzania has high fertility, low contraceptive prevalence and low exclusive breastfeeding (EBF). The Lake Zone, including Mara and Kagera regions, leads the country in total fertility; use of the lactational amenorrhea method (LAM) is negligible. This pre-/post-study explored the effects of a multi-level facility and community intervention (service delivery support, community engagement, media and LAM tracking) to integrate maternal and infant nutrition and postpartum family planning (FP) within existing health contacts. Mixed methods were used, including service statistics, exit interviews, patient-tracking tools for community health workers, client self-tracking tools, supervision data, focus group discussions and in-depth interviews. Results are presented using the reach, effectiveness, adoption, implementation and maintenance (RE-AIM) implementation science framework. The intervention reached primarily the second and fourth wealth quintiles, increased FP and EBF at six weeks postpartum. LAM was very acceptable, provided an entry point for FP conversations and for addressing misconceptions, and reinforced EBF practices. Partners felt encouraged to support spousal nutrition, breastfeeding and FP. Higher adoption in Kagera may be influenced by performance-based financing. The intervention was implemented with generally high fidelity. Maintenance data reflected stakeholder interest in continuing the intervention. A multi-level facility and community intervention was feasible to implement and likely contributed to improved EBF practices and FP uptake, including LAM use.

## 1. Introduction

Global evidence has demonstrated that short birth intervals are associated with adverse maternal and infant health outcomes. Studies have found that children born less than two years after their older sibling are more likely to suffer from poor birth outcomes, such as being small for gestational age, preterm birth, stunting or even premature death in the first year of life compared to children conceived at least 36–47 months apart [1,2]. The World Health Organization recommends waiting at least two years after the birth of a child before beginning the next pregnancy [3]. Early initiation of breastfeeding within one hour after childbirth, with skin-to-skin contact between the mother and the newborn, exclusive breastfeeding (EBF) for six months, and introduction of complementary foods at six months with continued breastfeeding are globally recommended practices [4]. However, timely postpartum contraceptive uptake, and early initiation and EBF for the first six months have posed a challenge in many low- and middle-income countries. Postpartum family planning (PPFP) and maternal, infant and young child nutrition (MIYCN) are closely intertwined and can be linked to strengthen provision of comprehensive care for women and children. One of the critical links is the lactational amenorrhea method (LAM) of family planning (FP), which provides an opportunity to synergistically reinforce both voluntary PPFP and MIYCN practices. LAM is a PPFP method that is over 98% effective [5] and requires adherence to three criteria: EBF, amenorrhea and baby less than six months of age.

Tanzania has a total fertility rate (TFR) of 5.2, high unmet need for contraception (22%), and a low modern contraceptive prevalence rate (CPR) of 32% [6]. Additionally, only about 50% of women initiate breastfeeding within one hour of giving birth and EBF is 27% among infants 4–5 months [6]. The Lake Zone, which is the location of the Kagera and Mara regions, leads the country in TFR at 6.4. Unmet need for FP in Kagera, at 22%, aligns with the national rate, while in Mara, it is higher at 34%. In Kagera, CPR is 39% while in Mara it is 29%. LAM use among all married women, regardless of postpartum status, is very low in both regions, at 0.5% in Kagera and 1.2% in Mara. Breastfeeding initiation within 1 hour of birth in Kagera is above the national average at 55.7% and much lower at 30% in Mara. Stunting in Kagera is high at 42% while in Mara it is 29% [6]. 

Maternal and child health and nutrition are key priorities for the Government of Tanzania, and several country strategies provide an impetus for integrated nutrition and FP programming. The government has set targets in the national reproductive, maternal, newborn, child and adolescent health (RMNCAH) strategy 2016–2020 (“One Plan II”) [7] to reduce stunting from 35% to 22% and is striving to improve EBF until six months of age from 50% to 80%, reduce TFR from 5.2 to 5.0, and increase modern CPR from 27% to 45%. The same national RMNCAH strategy highlights the need for higher coverage of a robust set of postnatal interventions, including immediate and EBF, complementary feeding, child spacing and the importance of maternal nutrition during lactation [7]. Community health workers (CHWs) are expected to be knowledgeable on the importance of EBF and discuss contraceptive options, including LAM [7]. Additionally, the Government of Tanzania’s National Multisectoral Nutrition Action Plan (NMNAP) [8] has prioritized scale-up of maternal, infant, young child and adolescent nutrition specific interventions (breastfeeding, complementary feeding) and aims to increase coverage of nutrition-sensitive interventions, across six development sectors, including health. Both the One Plan II and NMNAP strategies indicate the need for a multisectoral integrated approach to nutrition and FP. Furthermore, one key output of NMNAP is that “communities regularly use quality family planning prevention services” [8]. 

Program experience from other countries demonstrates that integrated PPFP and MIYCN service delivery is feasible and has led to improvements in service provision and health practices [9,10,11,12]. However, challenges remain around LAM uptake, adherence to the criteria among those who opt to use LAM, and timely transition from LAM to another modern method, as noted in studies in multiple countries [12,13,14,15].

This study was designed to assess the results of a multi-level community and facility intervention to improve PPFP and infant nutrition practices (including LAM as a contraceptive option) in Tanzania. The intervention itself was designed to address findings of a formative study conducted prior to the intervention [16]. The intervention took a socioecological approach, with activities at individual, household, facility and community levels. Collaborative teams of MCSP and government staff conducted quarterly supervision visits to intervention sites to collect data, assess intervention fidelity and develop and review progress toward action plans developed together with facility staff. Key findings of the formative study and intervention components are described in Figure 1.

## 2. Materials and Methods

### 2.1. Overview

The study aimed to examine, through an implementation science lens, whether a multi-level facility and community intervention to integrate MIYCN and PPFP within existing health contacts contributes to improving the following:Implementation (feasibility, acceptability and fidelity);Service outcomes (equity, client-centeredness, FP uptake and MIYCN practices).

Implementation took place from July 2017, in three districts in Mara (Tarime, Rorya and Musoma) and two districts in Kagera (Ngara and Misenyi). These districts were purposively selected to include the two regions’ main cultural groups to capture variations in practices so that learning from these districts could be extrapolated to other parts of the regions. Across the five districts, the intervention covered a total of 24 health facilities (six health centers and 18 dispensaries) and 50 villages within the health facility catchment areas.

Study data were drawn from a variety of sources, with the aim of triangulating results across various data collection platforms (Table 1) covering the period July 2016–June 2017 (pre-intervention) and July 2017–June 2018 (post-intervention). 

### 2.2. Qualitative Methodologies

Qualitative methodologies included in-depth interviews with mothers of infants under one year of age including current LAM users, former LAM users who had transitioned to another modern FP method, former LAM users who had not transitioned to another modern FP method, current users of other FP methods who had not previously used LAM, FP non-users, and current users and non-users of other PPFP methods, grandmothers of identified infants, and facility-based service providers who offer maternal and newborn health and FP services at public health facilities in the catchment areas of the 24 study facilities. Adolescent mothers, 15–19 years of age, were included. The study team also conducted focus group discussions (FGDs) with fathers of infants less than one year of age, influential community members and CHWs from the catchment areas. 

Using purposive sampling, we identified a total of 236 respondents for in-depth-interviews and FGDs. Experienced local researchers conducted the interviews in Kiswahili. Areas explored included: exposure to and perceptions of each intervention component, reported social and behavioral changes related to the intervention, and recommendations for further strengthening the approach. All participants provided verbal informed consent prior to participating in interviews or FGDs. Interviews were audio recorded and notes taken to supplement the recordings. The local research team transcribed the recordings and translated transcriptions from Kiswahili to English. 

Key themes to guide the analysis were identified and elaborated into a coding structure. The coding structure was updated to include themes that emerged from the analysis. Key analysis themes included: knowledge and uptake of breastfeeding/EBF/LAM, observed personal and community changes in MIYCN and FP practices influenced by the intervention, experiences with use of LAM, transition to other FP methods post-LAM use, and opportunities for integrating LAM with other maternal and child health services. No participant identification information was included in the analysis. 

### 2.3. Quantitative Methodologies

Service statistics data on early initiation of breastfeeding, total number of FP users and PPFP use, including LAM within six weeks (by contraceptive method), were extracted from MOH facility registers during quarterly supportive supervision visits for the 1-year implementation period (July 2017–June 2018), as well as the same period of the previous year (July 2016–June 2017). Data-entry staff then entered them into a study database in Excel. The study team verified the data entries and cleaned any gaps and inconsistencies between hard copies and the database. The data manager pulled data on EBF practice from DHIS2 [17], a government data warehouse; these data are collected during postnatal care (PNC) visits at six weeks postpartum. Thirteen facilities (six from Mara and seven from Kagera) with less than 75% completeness (more than 25% of data not reported) in service data for each indicator for each month were excluded from the service statistics analysis. One-tailed t-tests were used to assess for statistical significance in FP and breastfeeding service numbers before and after the intervention. 

During supportive supervision visits, fidelity assessments were conducted where each intervention component was scored as 0 (not implemented at all), 1 (partially implemented) or 2 (nearly fully or fully implemented). These scores were then summarized for each of the 24 facilities and divided by the total possible (10) to calculate percentage fidelity. Fidelity was analyzed over time and by intervention component to assess trends. 

Finally, the supportive supervision teams conducted structured client exit interviews with 193 women with infants under one year of age attending study health facilities for any maternal or child health or FP service over the course of four rounds of supportive supervision at intervention sites. Teams administered exit interviews using a paper-based structured questionnaire and included questions about services received, recall of key MIYCN and FP counseling content, and satisfaction with services. The study team conducted descriptive analysis of these data. The questionnaire also included questions from the EquityTool (www.equitytool.org), which were used to determine the relative socioeconomic status of interviewees to describe the reach of intervention activities across wealth quintiles. Study supervision teams included study staff and regional and council health management team members.

The study team explored results using the RE-AIM (reach, effectiveness, adoption, implementation and maintenance) implementation science framework pioneered by Russell Glasgow and colleagues and designed to more comprehensively assess public health interventions [18]. Since its development, the framework has been applied to numerous public health and other intervention studies around the world [19]. We used the framework to present qualitative and quantitative data, including the equity profile of people who received the intervention (reach), the outcomes (effectiveness), contextual factors in each region that may have affected the intervention (adoption), fidelity (implementation) and elements of sustainability (maintenance) as described in Table 2.

## 3. Results

In this section we present results along each of the RE-AIM domains, prefaced by a summary of the characteristics of respondents.

### 3.1. Characteristics of Respondents 

Characteristics of client exit interview respondents are included in Table A1. Characteristics of qualitative study respondents in each region, including age, education level and occupation, are presented in Table A2. For respondents to the qualitative study, the largest proportion of mothers, fathers, CHWs, and influential community members had completed only primary school education and worked as farmers/pastoralists/fishermen. Among mothers overall, the largest proportion were over age 30, with a higher proportion of mothers over age 30 in Mara than Kagera. Most women were married and had between one and three children with substantially more mothers in Mara having more than six children. In line with the sampling approach, mothers had infants 0–5 months and 6–12 months of age, and included a mix of FP non-users, LAM users, former LAM users and other PPFP users. Among fathers, the largest proportion of respondents was over age 30 and had infants who were 6–12 months of age. The majority of CHWs had completed primary school education, with a fairly even split between men and women. Facility service providers were largely nurse/midwives and more were female than male in both regions. More men than women comprised the sample of community influential people. 

### 3.2. Results per RE-AIM Implementation Research Framework Domains

#### 3.2.1. Reach

Reach analysis was conducted to assess whether the intervention was equitable; the analysis compared the wealth distribution of the population using the study facilities to the wealth quintile distribution of the regions as shown in the 2015–2016 Demographic and Health Survey (DHS) [6]. According to client exit interview data (*n* = 193), the largest proportion of users of health facilities came from the second lowest wealth quintile (Q2) followed by the second highest (Q4). Forty-one percent (41%) of the overall sample came from the two lowest quintiles (Q1 + Q2), 50% came from Q3 and Q4, and 9% from Q5. See Figure 2 wealth quintile distribution overall and by region. 

Generally, the proportion of health service users falling into the two lower wealth quintiles in the study areas was similar to the national distribution, reflecting equitable reach. In Mara, the proportion of users falling into Q1 was lower than in the recent DHS, but Q1 + Q2 combined were similar; another difference was that in the study area, more fell into Q4 and fewer into Q5 as compared to the DHS. In Kagera, the distribution was generally similar to the recent DHS; while Q1 + Q2 combined was similar, the proportion in Q3 was smaller and Q4 larger compared to the DHS. 

#### 3.2.2. Effectiveness

Effectiveness measures assessed service and behavioral outcomes for FP and MIYCN. Figure 3 and Figure 4 illustrate changes in service indicators comparing the intervention period to the same period of the previous year, as compared to number of first antenatal care visits (ANC1) and facility births. As shown in the figures, there were improvements across all indicators. 

##### MIYCN Service Outcomes

Service-delivery data indicate early initiation of breastfeeding (EIBF)—within 1 hour of birth—increased in both regions, demonstrated by the proportion of health facility-reported births with EIBF increasing from 97% to 102% (5% increase) in Kagera and 104% to 136% (31% increase) in Mara; however, the increase was only significant in the latter.

Exclusive breastfeeding practices reported at six weeks postpartum increased significantly in both regions, exceeding the number of health facility births. Using ANC1 as a denominator, the proportion of women practicing EBF showed improvements in both regions, with a more substantial increase in Mara, which went from 65% to 100% (a 54% increase), than in Kagera, which went from 66% to 77% (a 17% increase). Further explanation on the rationale for use of ANC1 as a denominator is included in the methods and discussion sections, with raw numerical data presented in Figure A1. 

##### FP Service Outcomes

Total modern FP use increased in both regions, with a 37% increase in Kagera and 11% in Mara, the change being significant only in Mara. Among all modern FP users, less than 1% in Kagera and 3% in Mara used LAM during the year pre-implementation, whereas 11% in Kagera and 13% in Mara opted for LAM during the implementation period. Use of a modern FP method within six weeks postpartum increased significantly in both regions (by 123% in Kagera and 101% in Mara). As a proportion of births, PPFP use by six weeks increased from 22% to 31% in Kagera (a 45% increase) and from 51% to 58% in Mara (a 14% increase). Among FP users within six weeks postpartum, the contribution of LAM to the method mix increased 49 percentage points in Kagera and 21 in Mara, reaching above 60% after the intervention in both regions. Among mothers using the LAM self-tracking tool, 58.3% transitioned to another modern method by six months. Use of long-acting reversible contraceptives and permanent methods (LARC/PMs) of FP among all modern FP users decreased proportionally in both regions, going from 44% to 38% in Kagera and 55% to 49% in Mara as LAM use increased. In spite of these changes in the method mix, the absolute numbers of LARC/PM users remained relatively unchanged among all modern FP users and those within six weeks postpartum. Figure 4 shows changes in total FP and use by six weeks, along with the FP method mix.

##### Behavioral Outcomes: Clients and Family Members 

Mothers, fathers and CHWs had a positive perception of LAM, commonly known as “*breastfeeding for family planning method*,” which was considered to be simple, easy to use, with no side effects. Mothers mentioned that they felt that EBF met the nutritional needs of their infant in the first six months of life, contributed to mental development, better growth and less illness in their infants, and that there was more attention paid to their own nutrition during lactation. The perceived benefits of EBF motivated mothers to use the method, and some recommended it to their peers:


*When I gave birth to this [baby] I had a training that there is a family planning method which is exercised by breastfeeding only…. The first benefit on my side is that the man knows that you are breastfeeding so at least there is an increment in the issue of nutrition. He knows that the baby is depending on you a hundred percent so he makes sure he provides you with good satisfactory porridge. Another advantage lies with the child. He grows with good health, he doesn’t get sick often. For example this one has never been sick, apart from scant rashes which are normal. If you look at him he is quite different from the other two because this one has become active at early stages.*
Mother, LAM user, three-month-old infant, Kagera region


*I heard people talking about LAM and was impressed, it is good for the child, the child grows healthy with good brain, it prevents the child from getting sick. For my other children, who did not use LAM, they were sick frequently. But now I am using LAM, I can see it has improved the health status of my child. I will recommend LAM to other women since it is a very good method.*
Mother, LAM user, four-month-old child, Mara region

Among mothers interviewed, EBF was the most commonly mentioned LAM criterion, followed by the infant being younger than six months of age. The LAM criterion of amenorrhea was not well remembered, and the concept of linking introduction of foods and liquids with transition to another modern FP method was generally not understood by mothers. LAM was seen to reinforce EBF practices, as mothers often equated EBF to LAM. As noted previously, 58% of women who were self-tracking their LAM use transitioned to another modern FP method by six months. 

Family members were also reported to be providing more support for women to exclusively breastfeed, including husbands reminding mothers to frequently breastfeed and mothers-in-law assisting with childcare while the woman works. Take the following as an example:


*[My husband] told me to breastfeed [the baby] for six months, he will be doing everything to make sure I get enough milk to breastfeed him.*
Mother, LAM user, three-month-old infant, Mara region


*My mother-in-law is supportive, she would come with me to the farm and carry the child while I work and when he cries, I get to breastfeed him.*
Mother, former LAM user, seven-month-old infant, Mara region

Providers also noted changes in women’s response to LAM:


*Yes, there is a difference which we can observe since we started to sensitize mothers on joining LAM, at least there is a huge response. The mothers have received it well, and after knowing the benefits of LAM, many of them want it and when you counsel them concerning LAM, they tell you, I will keep on breastfeeding my baby for six months, I won’t give him anything else.*
Health-care provider, Kagera

Health facility managers also noted the increased use of LAM, support from fathers for the method and better understanding of EBF in the community: 


*LAM uptake was low when the intervention was introduced… there was resistance… but, now we are seeing more women using LAM and husbands supporting their partners using LAM. They are seeing the benefit of LAM especially with regards to the health of the child. LAM has been instrumental in reinforcing exclusive breastfeeding in the community.*
Member district health management team

##### Behavioral Outcomes: Providers

Providers reported changes in how they provided counseling during antenatal, maternity and postnatal visits, with greater emphasis on maternal and infant nutritional aspects, including EBF and FP: 


*My perception has changed because I was wondering if LAM could be possible for all, but after being trained about how it works, I could see that it is effective as long as the three criteria have not changed…. It has changed the way I provide services. There are some things I did not know but now I know LAM and the criteria and now I focus on nutrition counseling since use of LAM depends on the ability of the mother to exclusively breastfeed.*
Health-care provider, Mara


*The biggest thing that enables them to accomplish the transition to another family planning method after LAM is the benefit she sees in the LAM program and the constant follow-up. It has become easy to encourage her to use another modern family planning method.*
Health-care provider, Mara

### 3.3. Adoption

Mara and Kagera regions are situated in the Lake Zone, an area that the Ministry of Health has prioritized due to high levels of maternal and newborn mortality [7]. While there are variations in the nutrition and FP indicators, these two settings have background characteristics that demonstrate similar nutrition and FP needs. There were differences between the regions with regard to complementary inputs from other projects and initiatives. Kagera region had additional nutrition-related and results-based financing program support, which included validation of register data, and may have resulted in better documentation compared to Mara. From the client exit-interview data, the coverage and performance of CHWs also appeared to be higher in Kagera than Mara. 

### 3.4. Implementation

Implementation measures draw from fidelity data, client exit interviews and qualitative data. Mean fidelity scores and client exit interview scores for each of the intervention components are shown in Table 3.

Fidelity of the intervention averaged 80% across the two regions, with Kagera scoring higher than Mara. Although fidelity in Mara started out lower than in Kagera, comparing the first and fourth quarters of implementation, there was an increase in the overall fidelity score in both regions, and for each of the intervention components, except the community engagement component, which remained around 50%. Notably, in Mara, fidelity of the self-tracking, CHW tracking and health-facility components experienced a decline between the third and fourth quarters of implementation. In both regions, fidelity to the community engagement component of the intervention was low. Exit interviews demonstrate the same with around only 11% of women reporting that they attended a community meeting where EBF and FP after childbirth and maternal nutrition were discussed. The LAM song fidelity scores were slightly lower in Mara (74%) than Kagera (84%); however, among exit-interview respondents, 65% in Mara and 51% in Kagera indicated they had heard something about LAM on the radio.

From in-depth interviews, women who heard the LAM song indicated it was beneficial for their understanding of the LAM criteria and the importance of breastfeeding; however, they did not report that it served as a source of information on *transitioning* from LAM to another modern FP method. 


*I heard the song, it helped me know the conditions of LAM, to breastfeed the baby frequently and for six months, it was very helpful and they should continue to air it.*
Mother, LAM user, six month old infant, Mara

During FGDs, fathers and influential community members discussed having heard the song, including at “evening joints”. The LAM song was positively perceived by men and considered informative.


*From the song, there is something that I have learned, that a child needs to take his mother’s milk only without mixing with anything like water or soup…. The words of the song say that the mother has eaten enough food so that she gets enough [breast]milk. Now, when I listened to the message, I understood very well that mothers need to eat food with vitamins. Now, we have to stop practicing our old ways, that if we eat, for example, hare meat we [the mothers] can’t get [breast]milk. I can see this song is very good together with its music.*
Influential community member, Mara

Mothers and fathers noted men’s increased support for breastfeeding and maternal nutrition. Men indicated they learned that infants should only receive breastmilk during the first six months and that mothers should eat nutrient-rich foods during lactation. Several women reported how their husbands supported them when practicing LAM, including ensuring that their nutritional needs were taken care of. They discussed their needs with their husbands and shared the LAM counselling they received from the CHWs and health-care providers:


*I discussed with my husband the need to make sure that we space and plan for our children, discussed about nutritional needs during pregnancy and when breastfeeding… Shared the LAM counseling I received from the community health worker with my husband and he has been helping with the use of the LAM card.*
Mother, LAM user, 4-month-old infant, Kagera Region

### 3.5. Maintenance

Maintenance of the intervention was explored by assessing fidelity scores over time, as well as qualitative data regarding sustainability of the intervention and suggestions regarding next steps. Fidelity data indicate that the implementation of intervention activities was sustained over the 1-year implementation period (Table 4). By the second supervision visit, about six months into the intervention, both regions were scoring above 80% overall. By component, fidelity scores were 85% or higher for self-tracking, CHW tracking and integrated service delivery at a health facility; they stayed that way throughout implementation. In both regions, the community engagement component was around 50% throughout implementation. 

Qualitative data show that, generally, regional and council health team managers in both regions felt the intervention could and should be continued. The following was noted by a manager in Kagera:


*I think this should spread, even in my local government and other centers. When we go to these mentorships and supportive supervision you can push it, but it is not in the important tools, which you can track their progress; but if this is there, it means we are widening the scope for the improvement of the mother’s health. We improve the health of the child and improve the prowess of the child.*
Manager, Kagera

Facility managers also had recommendations for adjustments to the approach to enhance the feasibility and effectiveness. One such recommendation was to provide additional support to CHWs to motivate improved performance: 


*The main actors who are the community health workers [should] be given incentives to implement this program more productively; as many of them work in conditions which they know they are not paid and do not have anything to facilitate them. They work in conditions which are demoralizing, but if this can be improved so they can work with confidence, it is possible for them to do much better.*
Facility manager, Kagera

Regional and council health team managers in Mara emphasized the importance of investing further in reaching husbands to enhance implementation and outcomes. Take the following as an example:


*Let’s direct our efforts to the men and give them priority as they have been complaining that the focus is on the women alone. I think it is time to get more manpower so as to focus on them and give them more education and enough knowledge so as they should also plan on how many children they would like to have, and at what time.*
Manager, Mara

## 4. Discussion

Our results indicate that PPFP and infant nutrition practices can be improved through integrated activities at the individual, household, facility and community levels. We organize the discussion around the RE-AIM domains. 

### 4.1. Reach

The intervention extended to pregnant and postpartum women from different socioeconomic backgrounds. An important element that needs to be emphasized is that mothers should increase the duration of breastfeeding (i.e., infants feed at the breast until both breasts are empty) and that mothers feed infants on demand, whenever infants demand it (day and night). In the formative phase of this study [16], while mothers reported breastfeeding, EBF was often disrupted due to early introduction of foods and liquids, such as cow milk, tea, porridge, juice and soda prior to six months of age. Notably, after the intervention, perceptions around breastmilk insufficiency changed, as health providers relayed that mothers were convinced that breastmilk alone was sufficient to nourish infants since breastfeeding is based on frequency (i.e., sufficient times breastfed, day and night, on demand) and duration (i.e., until both breasts are soft and empty), which stimulates breastmilk production—the “supply/demand” physiology of breastfeeding. The study also points to the potential for increasing male engagement and couple communication around FP and maternal and infant nutrition through an integrated multi-level approach. This is consistent with findings from Nyoni et al. that shows that male partner support helps women counteract pressures for premature mixed feeding [20]; our study offers implementation experience into programming this in real world rural settings. A recent study demonstrated an association between intimate partner violence and lower rates of EIBF and EBF [21], which is highly relevant for the Mara and Kagera regions, where, according to the DHSI 2015–2016, the percentage of ever-married women who experienced physical or sexual violence by their partner is 56.8% and 32.6% in Mara and Kagera [6], respectively. 

### 4.2. Adoption

In Mara, PPFP use within six weeks increased substantially, while total FP use, which includes the number of women using FP beyond six weeks postpartum increased only marginally. Adoption of FP use in Mara may be hampered due to entrenched gender inequities where the proportion of women who experience controlling behaviors from their partner is the highest in the country at 39.7% [6]. In Kagera, existence of a results-based financing program that included the number of new and return FP users may have had an additive effect across indicators. 

It should be noted that while service outcomes improved after the intervention, there are several remaining challenges hindering optimal behavioral uptake that require further attention. For example, the decrease in overall and postpartum use of highly effective LARCs and PMs, with attendant rise in LAM use, merits ensuring that counseling addresses women’s concerns about LARCs and PMs. Future efforts should ensure that health workers provide comprehensive counseling on the range of contraceptive options and that enthusiasm for LAM does not detract from women’s realization of their reproductive intentions over the longer term, including use of longer-acting modern FP methods, while still exclusively breastfeeding for six months postpartum. Furthermore, although breastfeeding indicators showed substantial improvements in both regions, key challenges to EBF remain, as reinforced by the qualitative results. For example, while some women reported bringing their infants with them to the field while working, return to work early after childbirth posed a challenge to maintain EBF. Additional focus on engaging communities to identify strategies for postponing resumption of work are needed, as this intervention’s scope was not sufficient for tackling that important barrier. More effort is also needed to improve complementary feeding practices in Tanzania, specifically around minimum dietary diversity and minimum meal frequency [22]. 

In both regions, there is a lack of understanding of postpartum return to fecundity, pregnancy risk, and importance of timely PPFP uptake and LAM transition to another contraceptive method. While knowledge of LAM appears to have improved, troubling problems with recall of the LAM criteria remained even after the intervention, especially for the criteria of amenorrhea and the baby being less than six months, with respondents using language conflating EBF and LAM. This likely hindered women’s use of the method and continuous pregnancy protection. In other contexts where MIYCN and FP interventions have been implemented, similar gaps in understanding of LAM, recall of the three criteria, and understanding of the importance of timely transition have also been identified [9,12]. Further exploration around cultural constructs regarding biology and risk, and testing of additional communication approaches may be needed. It is possible that the idea of three “criteria” may not resonate well and that we need to consider other ways to communicate cues for postpartum return to fecundity linked with introducing foods and liquids other than breastmilk, and PPFP transition linked with introduction of complementary feeding. The rate of transition from LAM to another modern method of FP among LAM self-trackers in this study was 58% by six months, reflecting a gap in timely transition among many LAM users. Other studies on LAM transition have noted similar issues. While this study demonstrates a possibly higher rate of transition than previous studies, this tendency of LAM adopters to fail to transition to another method when LAM is no longer viable remains a major challenge. For example, a study in Jordan revealed that only 48% of LAM users were using another modern FP method at one year postpartum [23]. A study in Egypt compared rates of transition between LAM users who were given emergency contraception along with LAM counseling, and those who were given LAM counseling only [24]. Rates of transition were 30.5% among the LAM and emergency contraception group and only 7.3% among the LAM-only group within or shortly after the first six months postpartum [24]. Results showing increased rates of transition when emergency contraception was offered provided rationale for the intended provision of emergency contraceptive pills along with LAM counseling in this study; however, our study results indicate that emergency contraception was generally not offered with LAM counseling as intended. It is quite possible that combined use of both a self-tracking tool and advanced distribution of emergency contraceptive pills in practice would have further magnified rates of transition in this study. In this study, we were unable to detect the true proportion of LAM transitioners, as we only had data from those who were using the tracking tools, but results suggest the tool does offer a behavioral cue to transition.

### 4.3. Implementation, Effectiveness and Maintenance

Implementation fidelity was generally high, due to efforts to train, supervise and mentor health workers. Strengthening the community engagement component may enhance effectiveness and equity in the future; a study in Kenya showed improved PPFP uptake with a community health strengthening intervention [25]. Community engagement will require more intensive, efforts, time and resources to support. At the end of the study, key stakeholders showed support for the intervention, indicating there is potential for continuation of the approach. In the 1-year period of intervention, we saw early indications of intervention maintenance as indicated by the continued high rates of implementation fidelity, which were sustained over time. We implemented this intervention in real world conditions and addressed some elements for successful scale-up, such as having engagement and governance of the intervention conducted jointly with the regional and council health management teams and facility in-charges, having a training package for community and facility-based implementers, and using client-centered tools. Moving from pilot to a broader scale, a systematic approach, such as that outlined in the Expandnet framework [26], is recommended. 

The self-tracking element of this intervention was informed by user-centered design, nonetheless, further refinements could provide additional benefit. There is growing interest in self-care approaches and products, including a new set of guidelines issued by World Health Organization [27]. LAM self-tracking tools are a form of fertility awareness monitoring. While self-tracking is not new [28], use of an instrument like the LAM tools in our study could potentially serve as an entry point for other fertility tracking, including use of digital tools. We are aware of only one digital tool specifically for postpartum women, which was developed by WHO [29]. Whether there is an appetite for a digital version of the LAM self-tracking tool used in this study remains unclear. A paper version, though inexpensive, may face limitations in terms of reproducibility in resource-constrained environments. Further research is needed to explore whether going digital would improve use of the intervention. 

### 4.4. Limitations

There may be limitations in generalizing these study findings to other settings, given the influence of unique sociocultural determinants at intervention sites. Measurement of EIBF proved to be a challenge through routinely collected data at health facilities, which may have affected data quality. At maternity wards, nurses often do not have time to document data collected immediately after childbirth; often, data are entered by other facility staff, and the staff frequently rotate, as noted in a recent study from Malawi [30]. Furthermore, data on EIBF reported through the national DHIS2 may include data from facility and home-based births, whereas birth data generally only include facility births, thus leading to coverage of EIBF in excess of the number of births reported. Exclusive breastfeeding was measured at six weeks rather than six months, which did not allow for quantitatively determining whether the interventions were effective for improving EBF at six months. Additionally, the six weeks PNC visit was likely not well documented in the DHIS2. After exploring with staff and managers at the study facilities why this might be, they reported that the report is often affected because there is a shortage of staff to compile and send data reports. ANC1 is almost universal in Tanzania—96% of women get at least one ANC visits. Therefore, the number of ANC1 visits was used as a proxy, to be a consistent measure that approximates the number of births, both facility and community. We had intended to disaggregate the total FP users by new and continuing, but, there are discrepancies in how the term “new FP users” is understood at facility level, and the variability left the indicator very difficult to interpret. Due to incomplete data, three Mara facilities and seven Kagera facilities had to be removed from the analysis of services statistics; having data from these sites would give a more complete picture. Strengthening the quality of routine data and documentation in health facilities will provide a more comprehensive picture of service outcomes and inform lessons learned; future efforts may want to apply a human-centered design approach to ensure that the flow of tools and key information is efficient and minimizes the burden on providers. 

It should also be noted that we applied the RE-AIM framework retrospectively; had we prospectively applied the framework as the intervention was designed and implemented, we may have been able to more comprehensively captured results across the framework’s domains. We were unable to rigorously track the maintenance component of RE-AIM, given that this study was a proof-of-concept study; however, we are able to point to elements of the approach that could facilitate future maintenance if the approach were to be adapted and expanded. Another limitation is that the fidelity measure scoring was done by the program supervision team—we tried to counter potential bias by having clear criteria for the scores; nevertheless, the measures were still subjective.

## 5. Conclusions

The study demonstrated that enthusiasm exists for LAM as a contraceptive option within the FP method mix, and that use of LAM benefited EIBF, EBF and couple communication. While FP indicators also improved, qualitative data reinforce the fact that further work is needed to improve our understanding of postpartum pregnancy risk and timely transition from LAM to another modern FP method. Use of the RE-AIM framework allowed for exploration of various implementation domains, which provide concrete insights into the complexities of programming, providing actionable learning for future programming.

## Figures and Tables

**Figure 1 ijerph-18-04105-f001:**
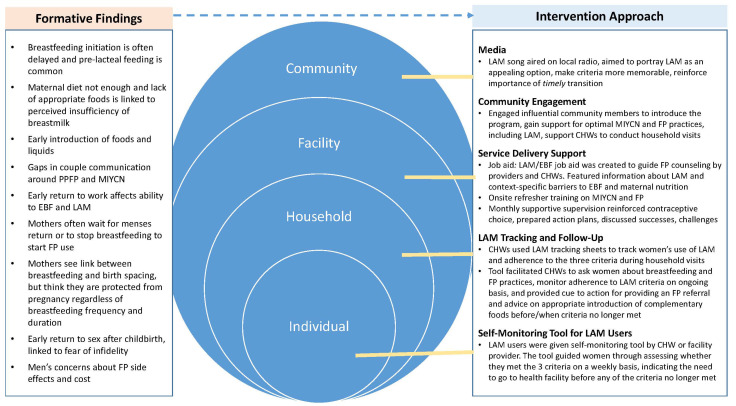
Formative study findings and intervention components.

**Figure 2 ijerph-18-04105-f002:**
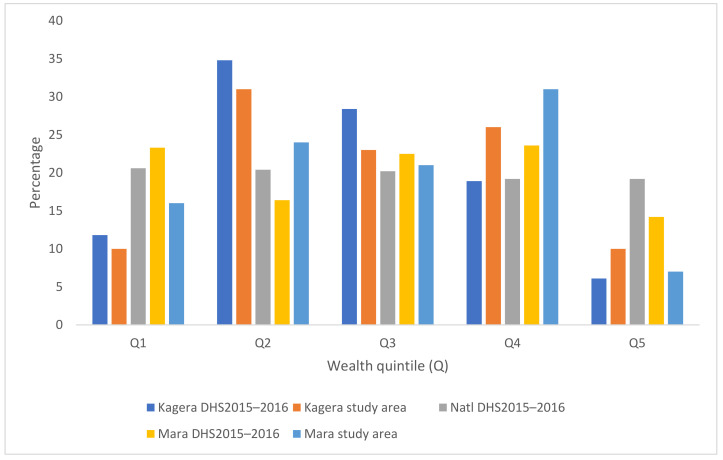
Wealth quintile distribution in study areas.

**Figure 3 ijerph-18-04105-f003:**
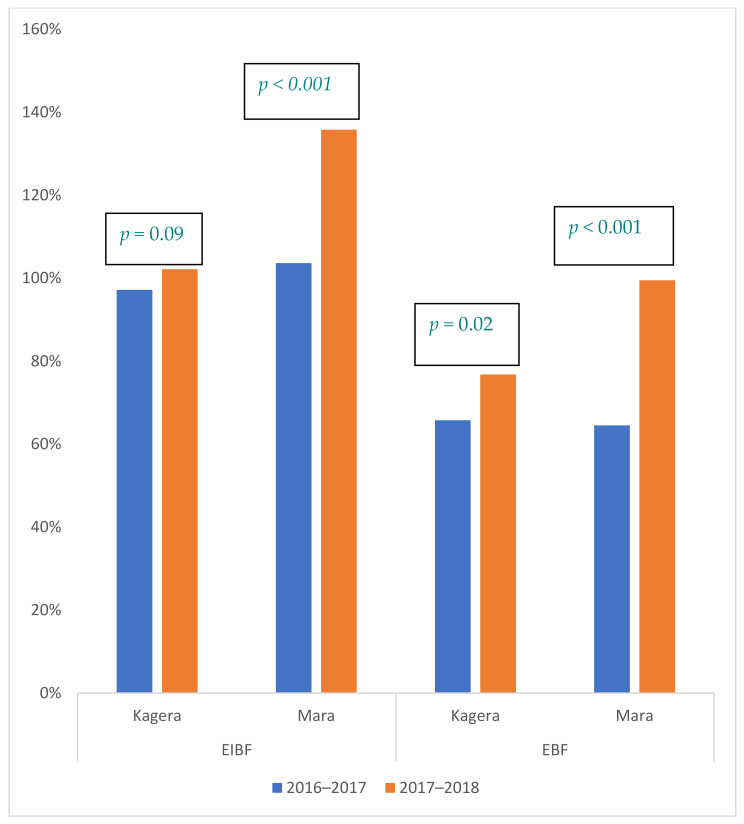
Percentage of women who practiced early initiation of breastfeeding (EIBF) * and percentage of women who practiced EBF ** at six weeks postpartum. * EIBF: Data are reported for all women, even if the baby was born outside the facility (e.g., born before arrival) and the mother came for immediate PNC. ** EBF: Number of ANC1 clients is the denominator. The EBF number reported far exceeds the number of births, and PNC data are not complete; therefore ANC1 data are used as a constant measuring stick to assess change over time.

**Figure 4 ijerph-18-04105-f004:**
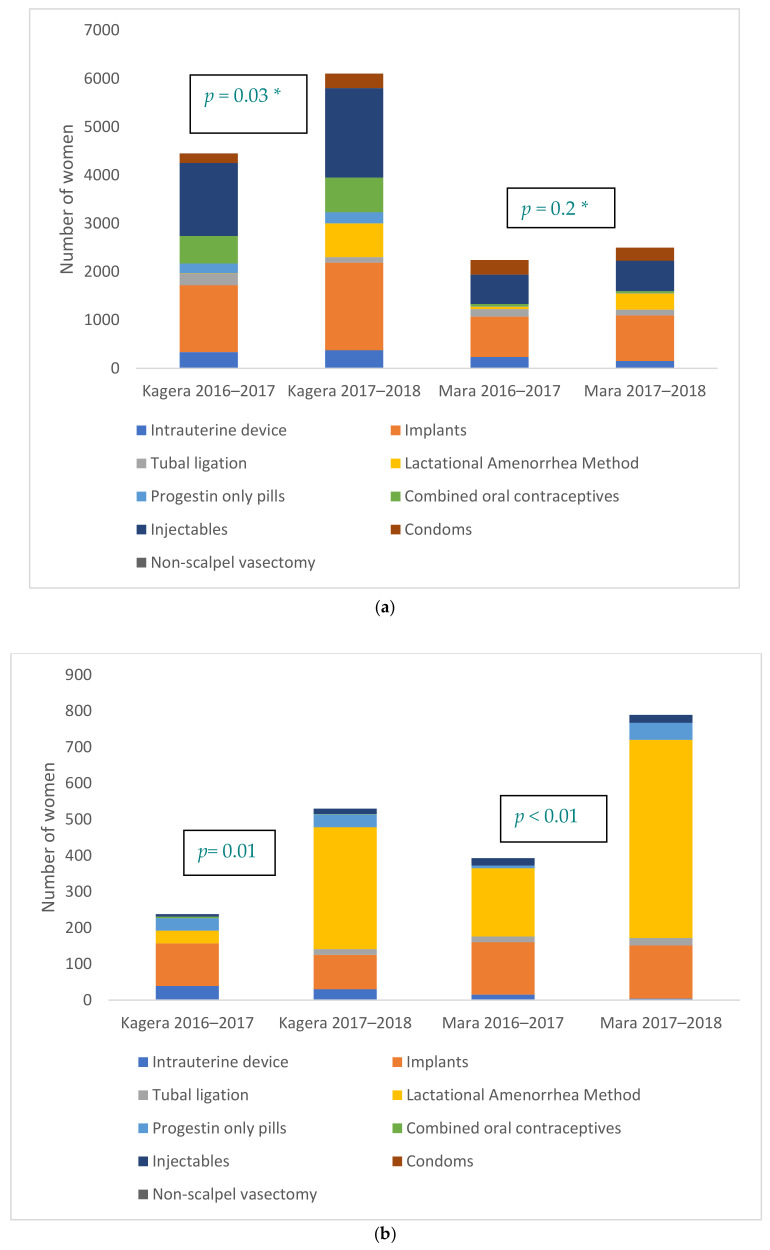
(**a**) Total FP users, by method, Mara and Kagera (* based on comparison of total number of users). (**b**) Number of women initiating FP by six weeks postpartum, by method, Mara and Kagera.

**Table 1 ijerph-18-04105-t001:** Data sources and measures.

Data Source	Measures/Content
MOH HMIS facility labor and delivery register	Breastfeeding initiation within 1 h of childbirth
MOH HMIS facility FP register	Total FP use by method
MOH HMIS facility PPFP register	PPFP by six weeks
MOH national HMIS postnatal data	Reported EBF by six weeks postpartum facility visit
Client exit interviews conducted during supervision *	Equity, perceived quality of care, timely transition
LAM tracking tools *	LAM use and transition
Fidelity checklist *	Performance of each intervention activity at each facility and surrounding communities over time
Qualitative in-depth interviews and focus group discussions with mothers, fathers, influential people, health facility managers, council health management teams, regional health management teams, national program managers	Formative: Practices, perceptions, barriers and facilitators for optimal MIYCN and FP practices Endline: Experience with intervention, reported changes, challenges and recommendations

Abbreviations in table: MOH, Ministry of Health; HMIS, health management information system; FP, family planning; PPFP, postpartum family planning; EBF, exclusive breastfeeding for six months; LAM, lactational amenorrhea method; MIYCN, maternal, infant and young child nutrition. * Introduced by the program/study.

**Table 2 ijerph-18-04105-t002:** Reach, effectiveness, adoption, implementation and maintenance (RE-AIM) domains, definitions and measures.

Domain	Definition	Measures/Content
**Reach**	Individual level measure; characteristics of those who receive or are affected by the policy or program	Wealth quintile of people reached
**Effectiveness**	Assessing positive and negative consequences (behavior, quality of life, satisfaction, physiologic endpoints, health and survival)	Number of total FP users (new + continuing), disaggregated by method Number and percentage ^1^ of PPFP users within six weeks, disaggregated by method Number and percentage ^1^ of women initiating breastfeeding within 1 h ^2^ Number and percentage ^3^ of women who reported to have practiced EBF up to six weeks postpartum Proportion of reported LAM users transitioning to another modern method of FP
**Adoption**	Proportion and representativeness of settings that adopt a policy or program; changes over time: differences in region, users/non-users, active agents versus non-active	Contextual factors in each region that may have affected implementation
**Implementation**	The extent to which a program is delivered as intended Individual level (e.g., adherence) and program level (staff delivery) Fidelity	Fidelity score Percentage exiting maternal and child health clients with infants less than one year reporting receiving intervention components Themes related to fidelity emerging from FGDs, including EBF and fidelity in use of LAM from client perspective
**Maintenance**	The extent to which a health promotion practice becomes routine and part of culture and norms of an organization	Trends in fidelity scores over time Feedback on desire to continue intervention, feasibility, recommendations for future efforts from key stakeholders

^1^ Number of births reported in DHIS2 is the denominator. This includes facility and non-facility births captured in the facility register and entered in national DHIS2. It is not disaggregated by place of birth. ^2^ Early initiation of breastfeeding data is reported for all women even if the baby was born outside the facility (e.g., born before arrival) and she came for immediate PNC. ^3^ Number of first antenatal care visits (ANC1) clients is the denominator. The EBF number reported far exceeds the number of births, and PNC data are not complete; therefore, ANC1 data are used as a constant measuring stick to assess change over time. ANC1 is almost universal in Tanzania.

**Table 3 ijerph-18-04105-t003:** Intervention fidelity and exposure across intervention components.

		LAM Song	Self-Tracking	CHW Tracking	Facility	Community Engagement
**Fidelity Score (Mean) *n* = 24**	Kagera	84%	98%	94%	98%	51%
	Mara	74%	85%	85%	85%	49%
**Exit Interview** (*n* = 193)	Kagera	51% had heard something about LAM on the radio	91% who opted for LAM were given a tracking sheet	68% who opted for LAM said CHW followed up to see the tracking sheet 43% received a visit from CHW after baby was born	65% were counseled on FP while at the health facility 25% who opted for LAM said they were given condoms and/or emergency contraception 70% women said they were given advice about feeding their child after six months 58% said they were advised on LAM	12% attended any community meetings where EBF and FP after childbirth were discussed
	Mara	65% had heard something about LAM on the radio	87% who opted for LAM were given a tracking sheet	25% who opted for LAM said CHW followed up to see the tracking sheet 20% received a visit from CHW after baby was born	63% women were counseled on FP while at the health facility 21% who opted for LAM said they were given condoms and/or emergency contraception 54% said they were given advice about feeding their child after six months 61% said they were advised on LAM	11% attended any community meetings where EBF and FP after childbirth were discussed

CHW, community health worker.

**Table 4 ijerph-18-04105-t004:** Mean fidelity score across intervention components over time, by region.

Region	Supervision Visit	LAM Song	Self-Tracking	CHW Tracking	Health Facility	Community Engagement	Average
Kagera	Supervision1	70%	93%	80%	93%	53%	78%
	Supervision2	93%	100%	100%	100%	50%	89%
	Supervision3	87%	100%	100%	100%	50%	87%
	Supervision4	87%	97%	97%	97%	50%	85%
Mara	Supervision1	56%	56%	50%	56%	44%	52%
	Supervision2	67%	94%	94%	100%	50%	81%
	Supervision3	72%	100%	100%	100%	50%	84%
	Supervision4	100%	89%	94%	83%	50%	83%

## Data Availability

The data presented in this study are available upon request from the corresponding author. The data are not publicly available, as quantitative data are used with the permission of the Tanzania Ministry of Health and qualitative data files may contain text that directly or indirectly identify study participants.

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
