# Peer review of "Implementing a Novel Facility-Community Intervention for Strengthening Integration of Infant Nutrition and Family Planning in Mara and Kagera, Tanzania"

_ijerph, 2021, doi:10.3390/ijerph18084105_

Round 1
Reviewer 1 Report
Implementing a novel facility-community intervention for strengthening integration of infant nutrition and family planning in Mara and Kagera, Tanzania
REVIEWER´S COMMENTS
GENERAL
This quasi-experimental study (pre / post intervention) has involved the implementation of a large cluster of activities, resources and efforts to explore the effects of a multi-level facility and community intervention to integrate maternal and infant nutrition and postpartum family planning (FP) within existing health contacts. However, I believe that the information and data that have been managed should be clarified for a better understanding of the reader.
SPECIFICS
Title
Integration of infant nutrition or perhaps the authors meant increased BF rates
Abstract.
Resources used to obtain the information are described here. It should be described in the abstract which interventions have been carried out to achieve the results described with this methodology. Could it be "multi-level training interventions in the community?
Introduction
The authors say that "The intervention itself was designed to address findings of a formative study conducted prior to the intervention" but then, this study does not carry out such an intervention? Does it only describe the current situation? If so, then it is not a pre-post study but a descriptive study
Materials and Methods
In general, I think this section should contain the type of work that has been done, describe in depth, and perhaps more independently, the sample and how it has been selected, state and regional sources of information (funding program in Kagera), interventions carried out. Better describe the intervention approach or intervention component, especially the Fidelity. Specify the temporality When have you done? (monitoring visits?) and finally the outcome measures how have we measured the changes? Where the authors have defined what has been done with the RE-AIM domains. Many of these data are interpreted throughout the document but I think it would be much better if the description were made here.
Table 1. The "table abbreviations" paragraph should be written in a smaller font size. What is the program/study? If the interventions that have been carried out are those that are announced in the table, in the measures of "Qualitative in-depth interviews and focus group discussions" it is appropriate to describe more exhaustively the training activities in this section. Clarify the information offered to participants, the intervention components, since this is the central and most important part of the study, are the activities implemented to achieve change
The term "DHIS2" first mentioned in this table has not been described
Line 151. What are the "quarterly supportive supervision visits"? In line 399 it is quoted that the "the second supervision visit, about six months into the intervention" in what time are the others? What interventions are carried out in each "supervision visit"?
Clarify in this section of material and methods, the temporality of the interventions. Line 152. Is the period of implementation of the activities carried out in this study is the July 2017-June 2018 period? Are the scope and effects that have occurred as a result of this implementation compared to the previous data for the period July 2016-June 2017? Line 167 cites "four rounds of supportive supervision".
Line 160. Where is this rating or score used? The rest of the document does not use this "fidelity assessments" scale. However, in Table 3, line 351, the "Mean fidelity scores" appears in percentages.
Line 174. Describe what is "MCSP program"
Line 185. Table 2. Clarify where the concepts are explained with superscript "1" and "2" since in the table it says, "disaggregated by method" but in the clarification below the table in both terms it indicates that "It is not disaggregated". If they match just leave a clarification.
Result
Paragraphs 3.2.2.1 and 3.2.2.2. It would be very useful if the authors provided quantitative information on the impact of interventions. That is, not only the increase in exclusive breastfeeding practices or FP but also to see if this increase is statistically significant, providing statistical measures of impact.
Line 246, line 250, line 251, 259-261. The increase in percentages does not match the description. Is this due to using the ANC1 or PNC as a denominator?
Behavioral outcomes: clients and family members. Line 271. Have the authors collected quantitative information from the results data they describe? For example line 294: "the most commonly remembered LAM criterion”, how much more?
Line 343. It may be useful to specify what this funding programme in Kagera consists of by describing it in the material and methods section, where the sample is described and its characteristics
Line 351. Is the "n" in "Fidelity Score (mean)" 236? Should it be reported as in "exit Interview"
The information between lines 353 and 365 repeats the contents of the table
Line 550. Here the authors specify that the "EBF data is collected at the six weeks PNC visit, and includes both facility and community births" and in Figure 3, line 231, it states that "Number of ANC1 clients is the denominator". However, in the line 235 and in the line 555 they say that "PNC data is not complete" and "The six weeks PNC visit was poorly documented in the DHIS2" then, really How has the prevalence of EBF been taken into account and the increases achieved after the interventions?
Line 628. In the title of the Figure, it is advisable that the authors do not put acronyms and if they do, independently describe each concept below. Now, some are, and some aren’t.
Discussion
In the section on "Adoption" as in the section on "Implementation, Effectiveness and Maintenance" the authors have expressed in excess the weaknesses or limitations of the study and again describe the results obtained but do not compare or discuss the results of other research and/or authors in these areas.
Limitations
Authors should rewrite this section to exclude information that should be included in the material and methods section (especially from line 548 to 576). They should be more concrete and specific, summarizing further the information regarding the limitations encountered in this work.
Conclusions
Line 586- 590 does not refer to conclusions of this study. It is advisable to rewrite this section in response only to the proposed objectives and the benefits obtained.
Line 591-594. I do not consider it to be part of the conclusions. This information could be cited in the discussion section if the authors so wish.
References
Check the bibliographic references that have been extracted from web pages since most do not contain the data from when this information was accessed.
In addition, there are errors in the wording of the references: vi, viii, xvii (author and access data), xxi, xxii, xxv and xxviii
Reviewer 2 Report
This study is very well executed and written up. I have very few qualms with it and most of these are minor in nature. Well done! Below, I articulate the few points that need to be addressed in my view prior to publication.
- Figure 1 is quite difficult to read. Can you replace it with a higher quality version?
- The study is thorough but would benefit from more context. The citations are relatively few and far between. Please integrate more findings from Tanzania specifically and Eastern and Southern Africa more generally.
Round 2
Reviewer 1 Report
I consider that the authors have made the suggested changes and the presentation of the research study has been improved.